# Small Ruminants and Its Use in Regenerative Medicine: Recent Works and Future Perspectives

**DOI:** 10.3390/biology10030249

**Published:** 2021-03-22

**Authors:** Rui Damásio Alvites, Mariana Vieira Branquinho, Ana Catarina Sousa, Bruna Lopes, Patrícia Sousa, Carla Mendonça, Luís Miguel Atayde, Ana Colette Maurício

**Affiliations:** 1Centro de Estudos de Ciência Animal (CECA), Instituto de Ciências, Tecnologias e Agroambiente (ICETA) da Universidade do Porto, Praça Gomes Teixeira, 4051-401 Porto, Portugal; ruialvites@hotmail.com (R.D.A.); m.esteves.vieira@gmail.com (M.V.B.); anacatarinasoaressousa@hotmail.com (A.C.S.); brunisabel95@gmail.com (B.L.); pfrfs_10@hotmail.com (P.S.); cmmendonca@icbas.up.pt (C.M.); ataydelm@gmail.com (L.M.A.); 2Departamento de Clínicas Veterinárias, Instituto de Ciências Biomédicas de Abel Salazar (ICBAS), Universidade do Porto (UP), Rua de Jorge Viterbo Ferreira, 4050-313 Porto, Portugal

**Keywords:** goat, sheep, small ruminants, animal models, regenerative medicine

## Abstract

**Simple Summary:**

Small ruminants such as sheep and goats have been increasingly used as animal models due to their dimensions, physiology and anatomy identical to those of humans. Their low costs, ease of accommodation, great longevity and easy handling make them advantageous animals to be used in a wide range of research work. Although there is already a lot of scientific literature describing these species, their use still lacks some standardization. The purpose of this review is to summarize the general principles related to the use of small ruminants as animal models for scientific research.

**Abstract:**

Medical and translational scientific research requires the use of animal models as an initial approach to the study of new therapies and treatments, but when the objective is an exploration of translational potentialities, classical models fail to adequately mimic problems in humans. Among the larger animal models that have been explored more intensely in recent decades, small ruminants, namely sheep and goats, have emerged as excellent options. The main advantages associated to the use of these animals in research works are related to their anatomy and dimensions, larger than conventional laboratory animals, but very similar to those of humans in most physiological systems, in addition to their low maintenance and feeding costs, tendency to be docile, long life expectancies and few ethical complications raised in society. The most obvious disadvantages are the significant differences in some systems such as the gastrointestinal, and the reduced amount of data that limits the comparison between works and the validation of the characterization essays. Despite everything, recently these species have been increasingly used as animal models for diseases in different systems, and the results obtained open doors for their more frequent and advantageous use in the future. The purpose of this review is to summarize the general principles related to the use of small ruminants as animal models, with a focus on regenerative medicine, to group the most relevant works and results published recently and to highlight the potentials for the near future in medical research.

## 1. Introduction

The use of small ruminants (e.g., mainly sheep and goats) in scientific research has been increasing significantly in recent years. Initially these species were used as animal models for human diseases and in research in animal nutrition and agriculture. More recently, adding to their initial functions, small ruminants have also started to be used in more complex studies of biotechnology, genetics and molecular biology, being essential sources of information in the fields of human and veterinary clinics, agriculture, anatomy and physiology and other fundamental sciences [1]. Although the scientific literature presents a multitude of works and projects involving small ruminants in all fields of research, the most significant advances and recurrent use of these animal models have been hampered by a difficulty in establishing standardized models of injury, disease and therapeutic protocols. There is a great variability in terms of the selected breeds, surgical interventions performed and treatments instituted that make it difficult to compare results between different works and more universal and unequivocal conclusions. Their benefits in research, for example in comparison with cattle, are instantly related to its smaller dimensions, ease of handling and low maintenance costs in reduced areas. When compared to rodents, the animal models most commonly used in any field of scientific research, small ruminants cannot compete with the fact that these animals breed rapidly and easily, allowing researchers to obtain entire generations of identical and genetically modified animals adapted to certain phenotypic profiles. The quantity and variety of reagents available to study their biology, genetic and immunogenic characteristics is also much higher. However, rodents become less desirable models in assays where thorough surgical interventions or large sample collections are required, and in these cases the great anatomical, physiological and immunological differences make it preferable to use models that are more complex and phylogenetically closer to humans, such as sheep or goats.

The potential of the small ruminants began to become evident when these animals were used as preclinical models in works of cardiac transplantation and in the application of cardiac valves and vascular stents [2], but these species have also proved to be useful in studies of reproductive cycles and improvement of artificial insemination and embryo transfer techniques, in addition to being the starting point for the use of revolutionary techniques such as cloning, gene transfer and general genetic engineering [3,4].

Traditionally, genetic selection in small ruminants was aimed at obtaining populations of animals with better productive performance, but it was essentially carried out by producers and hampered by small animal populations, planning and organization problems, poor animal identification, poor performance and pedigree registration. More recently, however, genetic selection techniques and new molecular tools have become more accessible and standardized and have also started to be applied to improve functional traits such as reproductive performance and disease resistance, allowing the selection of more targeted strains and for scientific research. This genetic selection is essentially based on cross- or multi-breed populations [5,6]. In the last decades, sheep and goats have also contributed to the development of genetic modification technologies. Although they have been used mainly in the field of genetic engineering to improve their characteristics and increase the production of milk, meat or wool, recently they have also been used as genetic models in biotechnology. The development of new genetic editing technologies that have made this technique more accessible and cheaper has further increased the interest in using these species in genetic bioengineering studies [7].

Goats and sheep are ungulated hoofed animals. Taxonomically they belong to the phylum Chordata, class Mammalia, order Artiodactyla (whose main characteristic is cloven hooves), suborder Ruminatia (animals that regurgitate and rechew their food) and family Bovidae (mammals that present compartmentalized forestomach, an even number of digits and horns). From a dietary standpoint, they are herbivorous animals, and metabolically their only source of glucose is gluconeogenesis. The Capra subfamily includes goats and sheep. The domestic goat belongs to the species *Capra aegagrus hircus*, having evolved from western Asian goats. The *Ovis* genera and subgenera are those that include the domestic sheep (*Ovis aries*) but also the European and Asian sheep species [1].

Goats (Figure 1) were one of the species that were first domesticated, around 10,000 years ago in western Asia [8]. Currently, there are more than 300 breeds of domestic goats exploited mainly to produce meat, milk, skin and hair, with wide variations in physical characteristics, dimensions and weight. In addition to being easily available commercially, goats are also sociable, curious, gentle, intelligent, clean, easy to transport and maintain and robust animals, making them desirable and convenient animal models [9,10]. Normally goats do not need particularly complex infrastructure to be housed and maintained, but these must always be adapted to the exploratory and social behavior shown by this species, particularly in situations that require long-term confinement [11]. Goats are used as animal models in works in different scientific fields, from nutrition, parasitology, immunology, infectious diseases, chemotherapy, psychology, physiology and reproductive medicine. This increase appears in parallel with the decline in the use of domestic animals such as the dog, not only because ruminants raise far fewer ethical restraints than pet animals, but also because some physiological characteristics for goats make them ideal models of surgical study and training. Goats are interesting animal models for vascular studies due to their anatomical characteristics that facilitate the exposure and catheterization of large blood vessels: a long neck with little adipose tissue that allows access to a large and easily reachable jugular veins and well-developed hindlimb musculature with little inguinal adipose tissue [12]. In comparison with the dog, several goat joints are easier to access, and the anatomical constitution of the subchondral bones and cartilage have more similarities with humans than other species such as rodents, dogs and sheep [13]. The metabolic rate and bone remodeling are similar to that of humans, and that is why the goat has been widely used in studies of bone, cartilage and ligament regeneration and repair. In addition, since their body dimensions are relatively large, they have also been used in studies with implantation of grafts, biomaterials and prostheses, although it is known that grafts are incorporated and revascularized more quickly than in humans [14]. Goats have proportionally large hearts, which allows them to be targets of complex surgical interventions in cardiology studies, being an animal model in atrial fibrillation [15]. Goats with congenital myotonia (fainting goats) are models for studying the same condition in humans [16]. Because of the susceptibility to the caprine arthritis-encephalitis virus, they are also used as an animal model of human chronic rheumatoid arthritis [17]. In addition, the virus also cross-reacts immunologically with HIV, allowing a greater understanding of this human virus [18]. In the field of reproduction, the goat is known for the occurrence of the mutation and associated polled intersex syndrome, which has helped to understand sexual differentiation in mammals [19]. The use of transgenic goats to produce a wide variety of biologically active recombinant proteins and antibodies is also common. The desired proteins are collected and isolated from milk, and as such the ideal transgenic animal is one that produces large quantities of milk and has relatively short generation times, and the goat satisfies both these criteria [20]. The first drug originated from genetically modified animals and approved by the European and United States drug agencies was precisely an anticoagulant drug produced from the milk of transgenic goats (ATryn^®^) [7].

Due to the frequency with which they develop adrenocortical neoplasms and malignant melanomas, goats have also been proposed as study models for these pathologies [21]. Other fields of exploration of the goat model include metabolic and genetic diseases, research on osteoporosis and therapeutic cell transplantation [22,23].

Despite its increasingly widespread use, proportionally the goat continues to be used less frequently than the sheep. This fact can be related to their behavioral characteristics. Because it is a much more interactive, dynamic and social species, goats establish effective connections with humans much more easily than sheep, also raising more ethical questions when selecting these animals for the interventions included in the research work. In addition, their inquisitive, exploratory and less gregarious behavior makes them much more likely to escape and less suitable for assays that require long-term restrictive confinements [10]. Finally, the fact that the genome of goats, unlike sheep, has not yet been fully and unambiguously sequenced, limits the use of this species for genetic studies [24].

The sheep (Figure 2) is more traditionally used as an animal model than the goat, being the preferred species for research work in a wide range of fields such as reproductive and fetal medicine, circadian rhythm characterization, the relationship between smell and behavior, metabolic and congenital diseases. In addition, there are also common models for applying orthopedic procedures, drug tests and implant testing [1]. They are easily available animals, with low feeding and maintenance costs and more easily accepted by society as animals for research purposes. In addition, they are docile animals, easy to handle and with low feeding and housing requirements. Their size is similar to those of the humans, allowing good reproducibility in both surgical interventions, sample collection and obtainment of imaging information. Their gregarious behavior and common distrust with the approach of humans requires some space available in their housing for free movement, which may be a limitation in some facilities. The equivalence of age between humans and sheep is also well defined, allowing this variable to be used in the translation of results between the two species [25]. Surgery and anesthesia equipment are similar in size and characteristics to those used in human medicine, not creating spatial and budgetary limitations as with other large models (e.g., cattle) [26]. The similarities between human and sheep lung structure and functions, namely in terms of respiratory rate, resistance, and air flow, make the sheep a good model for asthma studies [27] and also for other complex diseases with genetic origin such as cystic fibrosis [28]. Preterm and term lambs also present pulmonary structures identical to humans in the prenatal and neonatal phases, namely in terms of airway branching, composition of submucous glands and pulmonary oxidative system, allowing its use as a model of diseases such as respiratory distress syndrome in preterm infants and respiratory syncytial virus infection [29]. The composition, metabolism and bone remodeling process are identical to the human, and the long bones make them ideal for the application of implant systems and devices and for osteoporosis studies [30,31]. Sheep are also commonly used as cardiovascular models for applying artificial replacement devices [32] and also in studies of female fertility and pregnancy disorders [33,34]. Other relevant works in the field of cognition have allowed to realize that sheep can be good models of study in the mechanisms of decision-making, facial recognition and triggering emotions [35,36,37]. The genome of the domestic sheep has already been sequenced, unlike that of the goat. This sequence allows new techniques of genetic manipulation with increased applicability in studies of genetic engineering and biomedical research [38], for example creating models for studies of human genetic diseases like Huntington’s Disease [39].

See Table 1, Table 2 and Table 3 for physiological and reproductive parameters of both species.

Despite their advantages as animal models for most systems, the use of small ruminants as models of gastrointestinal diseases is limited due to their four-stomach system, which fundamentally alters the bioavailability and effectiveness of drugs administered orally. It is also difficult to fast ruminants, as their rumen can provide nutrients for long periods. In addition, special care is needed when prolonged fasting or long-term administration of non-steroidal anti-inflammatory drugs or antibiotics is required, which can lead to acidity of the abomasum and consequent ulceration, which can also be aggravated by high levels of stress resulting from manipulation and inadequate diet [40]. Thus, pain management and antibiotic administration must be adapted to the physiological needs of these animals. Due to its gastrointestinal characteristics and diet, feeding has an essential role in these species and a direct outcome in their immune response and microbiological control. Diet must always be well planned and adapted to the species, and to stressors acting on animals during studies where goats or sheep are involved [41]. Particularly in sheep, the fact that they are stoic animals can make pain assessment difficult, and although there are some systems for evaluating facial expression, its practical applicability is not always ideal [42].

## 2. Small Ruminants as Animal Models and Regenerative Medicine

### 2.1. Nervous System

The main advantages of using sheep as models of peripheral nerve regeneration are the nerve dimensions and regenerative pattern identical to that of humans [43,44,45]. Even histologically, the nerves are polyfascicular just like humans [46]. Most works involving sheep are where the median or facial nerve is used, with a translational application in orofacial medicine or in hand surgery [47,48]. The nerves of the hind limb have not yet been the subject of many studies in this species, and those that exist are directed to the sciatic nerve [49] and its branches [50]. Other nerves used include the radial, tibial and the common peroneal nerve [51,52]. In addition to the choice of more traditional techniques involving the use of autografts and allografts, new tissue engineering methods have already been developed using cell-based therapies and biodegradable scaffolds and grafts with promising results [51]. At the level of the central nervous system, an ovine model of spinal cord injury has been established [53], and the effectiveness of the injecting MSCs (mesenchymal stem cells) in reversing the degeneration of intervertebral discs has also been studied [54]. The administration of BM-MSC at the level of the annulus fibrosus or the nucleus pulposus of degenerated intervertebral discs led to an improvement in all indices of disc health, and an influence of the site of administration on the therapeutic efficacy of MSCs was investigated [55]. Furthermore, mesenchymal progenitor cells primed with pentosan polysulfate were able to promote a better structural organization and proteoglycan content, and reduce the signs of degeneration in microdissected discs [54]. Additionally, the sheep has also been used as a model of human neurological disorders [56]. The sheep’s relatively long life expectancy makes it a great candidate for studies of neurodegenerative diseases such as Alzheimer’s or Parkinson’s disease where development is slow and progressive and arises in later stages of life [57]. An ovine transgenic Huntington’s disease model was already established [58]. In this disease the sheep has been shown to be a model with potential since the sheep brain is identical to the human both in terms of dimensions and in the structural organization and location of the cerebral striatal nuclei and cortex. The differences with classic models such as rodents are much more marked. Similar brain dimensions and characteristics also allow imaging and electroencephalographic techniques to be easily used and their results to be compared in a translational manner [59]. Additionally, it is known that the brain size and percentage of cortical white matter in sheep is closer to that observed in humans than in other species such as rodents, which makes this species a good model for studies of stroke and vascular dementia as it is known that in larger brains a higher percentage of white matter can be affected, leading to more severe functional consequences [60,61]. Mesenchymal stem cells (MSCs) can also be used to prevent sequelae associated with hypoxic-schematic lesions such as those identified in ovine pre-term brain hypoxia-ischaemic-injuries [62]. Two different types of MSCs (amniotic fluid derived and placenta derived MSCs) were used in myelomeningocoele models, promoting motor function and preserving large neurons in the spinal cord, with the second cell type guaranteeing better results [63,64].

There are far fewer studies on the use of goats as an animal model of nervous system regeneration. In peripheral nerve research, biomaterials and autologous bone marrow mononuclear cells were used together to promote the regeneration of a defect in the peroneal nerve with promising results [65]. Some genetic mechanisms associated with the maintenance of myelin in mammals have also been discovered using non-transgenic goats [66]. The neurotoxic effects and peripheral neuropathy due to copper deficiency are also well described in the goat, which can be used as a model for the same clinical entity identified in man [67]. Finally, other groups also described the peripheral neuropathic effect associated with the consumption of Coyotillo fruit, a phenomenon that has also been recorded in man [68,69]. A caprine model of acute central cervical spinal cord injury syndrome combined with chronic injury was created [70], and goats were also used to study the effects of different types of laminectomies on spinal cord injury subsequent to acute spinal shorting [71] and the effects of adipose stem cells seeded on a radiolucent cage filler in spinal fusion [72]. Particularly in intervertebral disc disease, the application of a gelatin sponge rich in goat bone marrow MSCs (gBM-MSCs) and platelet-rich plasma promoted healing, with histological evidence between 3 to 12 weeks [73], and other studies on the use of MSCs and basement membrane molecules appear to have potential for cartilage regeneration and chondrogenesis of the nucleus pulposus [74].

### 2.2. Cardiovascular System

The sheep has often been used as a model for cardiovascular diseases since its cardiac anatomy is similar to that of humans and its anatomy allow easy access to the pulmonary and aortic valves. A technique for replacing the pulmonary valve with a resorbable synthetic graft was developed, which allowed the colonization of the structure by host cells to be observed with the presence of newly formed tissue without signs of calcification [75]. The sheep is a good model for studies of myocardial infarctions because it mimics the development of this pathology in humans: dimension of the infarcted region, maintenance of blood flow in the non-infarcted region, and absence of collateral blood flow in the infarcted region [76]. The implantation of cardiac MSCs has already been tested in the sheep by three routes: endomyocardial, intracoronary and intraperitoneal [77]. Cell therapies were used to treat acute myocardial infarction, and the inoculation of MSCs allowed to decrease the level of fibrosis and promote angiogenesis and cardiac function (probably through differentiation in an endothelial line) [78,79,80], and also remodeling the region adjacent to the infarction area after being transplanted [81]. The administration of different types of MSCs intraperitoneally led to their differentiation in Purkinje cells in fetal sheep heart [82]. Different models of vascular grafts have also been developed and tested on sheep in a preclinical context [83]. In addition to studies directly related to cardiac function, the application of MSCs has also enabled the development of blood vessels [84], and the intravenous implantation of mesenchymal precursor cells reduced systemic inflammation and endothelial changes [85].

The goat has already been used as an animal model of heart disease in multiple studies, even allowing for major advances in open heart surgery [86]. The dimensions of the goat heart are very similar to the heart of an adolescent human, which makes them a good model of heart failure by coronary artery ligation, just like in sheep [87]. Goats were also used to create models of persistent atrial fibrillation associated with chronic left atrial overload [88,89] and in transgenic animals [90], the development of intelligent artificial papillary muscles for surgical restoration in valvular disease [91] and for fetal heart surgery techniques [92]. When used in a left anterior descending coronary artery ligation model, gBM-MSCs in combination with a small intestinal submucosal film were able to prevent left ventricular chamber dilation, ensuring good cardiac function, collateral perfusion and myocardial contractile capacity [93].

A limitation in the use of small ruminants as models of cardiac injury is related to their gastrointestinal anatomy and thoracic contours, that are different from monogastric species, which may make it difficult to obtain echographic images and require the use of more invasive techniques [76].

### 2.3. Respiratory System

The sheep has high potential as an animal model for respiratory diseases since the anatomy and physiology of the respiratory system in this species is more similar to that of humans than rodents [94], making it possible to apply vaccinations, induce inhalation treatments, measure certain respiratory parameters and proceed to cannulation and frequent collection of large samples [94]. This model also contributed to the development of new specific treatments such as mechanical ventilation systems, vasodilators, extracorporeal membrane oxygenation and nebulized surfactants [95]. Due to its temperament, in the sheep model the evaluation of pulmonary mechanics can be done in an unrestrained manner, in animals that are not anesthetized or just slightly sedated [96]. Asthma is a disease that is difficult to mimic, and sheep have been used as a model for particle-induced disease along with dogs, with clear advantages over rodent models, particularly with regard to the inflammation of the lower respiratory tract. A sheep model of asthma triggered by dermatophagoides (house dust mite) has recently been developed [95]. By sharing patterns of bacterial infections with humans, sheep models also allow studies aimed at understanding the patterns of colonization of the airways and consequent pneumonia with the need for assisted ventilation [97]. As part of the investigation of lung tumors, the anatomopathological similarities between pulmonary diseases of spontaneous occurrence in sheep, such as ovine pulmonary adenocarcinoma, and bronchioloalveolar carcinomas in humans, have allowed advances in the understanding of these diseases, facilitating the monitoring of disease progression by radiographic and endoscopic methods and the testing of therapies that could only be applied to larger animal models such as radiofrequency ablation or inhalation of anti-tumor therapies [95,98]. To explore other respiratory diseases, sheep models have already been used in the studies of acute bronchial obstruction, regulation of surfactant proteins, infant respiratory distress syndrome using premature lambs and adult respiratory distress syndrome [95]. The use of MSCs in the treatment of pulmonary pathologies has already been explored in different studies involving sheep, with, for example, improved oxygenation and decreased pulmonary oedema in cases of bacterial pneumonia [99], decreased impact of endotoxins with attenuation of inflammation in acute respiratory distress syndrome [100,101,102] and attenuation of pulmonary microvascular hyperpermeability after smoke and hot air inhalation [103]. In a sheep model of endotoxemia treated with intrathecal implantation of BM-MSCs, easier management of acute respiratory distress, less inflammation and better histological parameters were observed [104]. Endoscopic transplantation of autologous lung-derived MSCs in sheep with emphysema ensured good cell retention, increased extracellular matrix content with particular distribution in the alveolar septum and peribronchiolar interstitium and, in general, signs of functional regeneration of lungs with emphysema [105].

With regard to respiratory diseases, goats have been largely exploited models to study tuberculosis [106], namely for the development of vaccines to be used both in large ruminants and in humans [107]. The main advantages of using goats as a study model in tuberculosis are their low maintenance and housing costs and, above all, the fact that they develop tuberculosis lesions and immune response identical to humans in the active phase of the disease [106]. The cytocompatibility of a decellularized goat-lung matrix with different types of cells also demonstrated the potential for its use as a scaffold for tissue engineering applications [108]. In a bronchopleural fistula induction model, after the implantation of gBM-MSCs, healing of the fistula was observed after 28 days, with the presence of a collagen matrix and proliferation of extraluminal fibroblasts [109].

### 2.4. Urology

Sheep have already been used as models of different urogenital pathologies, again due to their anatomical size being very similar to humans that facilitates surgical procedures, therapeutic application, and monitoring of regenerative progression by different imaging methods. Particularly in comparison to women, the urethra of female sheep is of equal length and the passage from the external body opening to the urethral opening is equally short, being good models for urethral catheterization and urological measurement [110]. As a substitute for the pig as a classic model, sheep were established as ideal models to study healing after partial nephrectomy [111]. Sheep were also used as a model to study the efficacy and safety of using cryotherapy to remove renal malignancies with preservation of the parenchyma and renal tubules [112]. The use of a collagen scaffold to promote tissue regeneration resulted in identical outcomes in a diseased bladder model and with healthy bladders, with good histological results and significant amounts of regenerated tissue involving all layers of tissue [113]. The replacement of a resected bladder segment by biologically inert patches also demonstrated bladder regeneration at the replaced segment, with the presence of fibrous tissue and the growth of new blood vessels in the different histological layers, without decreasing bladder capacity [114]. The sheep has been shown to be particularly appropriate as a surgical model to train trans-obturator and retropubic sling techniques, due to the ease of catheterization and the similarity of the cystoscopy techniques compared to that applied in humans, with the urethral meatuses in the same position, and a trans-obturator access being equivalent despite the smaller diameter [115]. Some preliminary studies also indicate that the intra-arterial application of autologous MSCs in postischemic kidneys promote successful engraftment at the level of renal tubules and glomeruli, although there is still some uncertainty on its reparative effect [116]. The sheep is even a model of haemodialysis treatment, in order to prevent the adverse effects of renal replacement therapy [117].

In goats, models have been developed for the study of urinary incontinence and infection, with the establishment of females as useful models of human female stress incontinence disorder and treatment with MSCs [118] and in males in order to better understand the relationship between androgen level and the severity of urinary tract infections [119]. Electrical stimulated graciloplasty as a treatment for incontinence was developed in male goats, allowing creation of an animal model of urethral pressure measurement [120], and the use of tissue engineered templates and subcutaneous pre-implantation in the promotion of ureteral reconstruction of the urethra with extensive defects has shown favorable results [121]. The production of a natural three-dimensional goat kidney scaffold has enabled advances in understanding the regeneration of this organ, and opened doors for greater availability of these organs for donation [122]. A technique of renal subtotal artery embolization has been used to create a stable mildly uremic model [123].

### 2.5. Ophthalmology

Small ruminants are not common models of ophthalmic disease, probably due to the evident anatomical differences between the eyes of these species and humans [124]. Their application in this field is often *post-mortem*, with the use of sheep and goats enucleated eyes for surgical training [125,126]. Even so, there are some studies in which sheep models were used for the study of glaucoma, exploring the relationship between the application of ocular steroids and the development of intra-ocular hypertension [127,128]. A method using biomaterial scaffolds and ovine corneal endothelial cell was used to develop the establishment and subculturing of corneal endothelial cell cultures, minimizing the loss of cell-to-cell contact and epithelial-to-mesenchymal transition, which can have deleterious effects during corneal transplants by reducing the capacity of the cells to form a mature and functional layer [129].

The goat was used as a model for studying the reconstruction of the corneal epithelium. For this purpose, corneal limbal stem cells were applied with positive reconstruction of the damaged corneal surface. This outcome could be explained by the anti-angiogenesis activity of these cells, thus reducing local inflammation [130]. Likewise, epidermal adult stem cells explanted and cultured from the skin of an adult goat were used to reverse a damaged corneal surface in a goat model with total limbal stem cell deficiency, and the differentiation into corneal epithelial cell in a corneal microenvironment and the ability to activate corneal genetic programs was confirmed [131].

### 2.6. Osteoarticular System

Sheep have been widely and successfully used as animal models to investigate orthopedic diseases, due to their similar body weight to humans and bone sizes ideal for the application of human orthopedic implants and prostheses [132]. Different disease models have been developed for research, as fractures, osteoporosis, bone-lengthening and osteoarthritis. In terms of bone remodeling, humans and sheep present a similar bone in-growth pattern into porous scaffolds, and considering bone composition, both species present comparable mineral compositions [14].

Goats have also been used as animal models for orthopedic research. However, due to their stronger character, long confinement periods associated with orthopedic lesions’ recovery can present a practical challenge for a study. Nonetheless, in high-temperature regions, goats are preferably applied as they are more tolerant to high humidity and temperature. Comparable bone remodeling rates and mineral compositions to humans have been described in this specie [14].

The application of tissue-engineering techniques and MSCs have been successfully used in both sheep and goats. Lesion models vary from cartilage defects, like the femoral condyle, trochlear groove and mandibular condyle, to bone defects [133,134,135]. Bone defects can be classified as non-critical or critical defects. Critical defects are classified as those who will not self-heal spontaneously, with no treatment and in a defined recovery period [136]. Critical bone defects are commonly applied in the ileum crest, being a non-weight-bearing bone with integrity to support larger defects [137]. Non-critical bone defects are those who can self-heal spontaneously, thus without compromising the bone intrinsic healing capacity, and can be applied in various bones, like the femur [138,139], tibia [140,141] and others, like the mandibular bone [142,143].

Furthermore, sheep and goat are reliable animal models of osteoporosis for preclinical and translational studies to humans. Ovariectomized animals are the most frequent disease model applied, and are associated with glucocorticoid therapy for induced osteoporosis [144].

Regarding dental regeneration, both sheep and goat are reasonable candidates as animal models, although not widely used for this field. Nonetheless, various studies have used bone grafting techniques associated with dental stem cell therapy, such as dental pulp stem cells [139,145,146].

### 2.7. Skin

Wound injuries caused by traumas, surgery and pathological conditions can affect the primary barrier against external microorganisms and dehydration, impairing the health and wellbeing of individuals. Wound healing is often associated with four phases, being hemostasis, inflammation, granulation/proliferation and tissue remodeling. The healing of such injuries is a complex process, commonly associated with scar tissue formation, infection, chronic wounds and dysfunctional tissue regeneration, as keloids or hypertrophic scars [147,148]. Wound care is challenging, with treatments aiming at short term healing, minimizing discomfort and pain, thus restoring the normal function of the tissue [149]. Sheep and goats have not been commonly used as skin animal models for wound healing research. However, they are good candidates due to their easy temperament and anatomical skin superficial area available for the creation of lesion models of different shapes and sizes. Some groups have investigated the regeneration potential of different treatment in these animal species, applying different gels, Platelet-Rich Plasm and other topical treatments [147,149,150,151,152].

Moreover, some research groups have been associating MSCs in wound healing experiments, with improved results, suggesting MSCs may accelerate wound regeneration and shortening the healing time process by contributing to the re-epithelization, vascularization and extracellular remodeling of the skin [147,153,154].

### 2.8. Reproductive System

Sheep have been applied in the research of pelvic floor dysfunctions, associated with age, hormonal changes, parturition and others. Pelvic floor dysfunction includes many disorders, like pelvic organ prolapse and stress urinary incontinence. Sheep vaginal models have been used in several different approaches, as the application of meshes to prevent organ prolapse, and laser treatments to improve collagen levels and structure, as collagen is one of the most prevalent component of pelvic floor tissues and responsible for their mechanical behavior [155,156,157].

Sheep are reasonable candidates for these studies, as their anatomical structure allows the implantation of several and larger vaginal meshes with longer recovery periods [158]. They also present a more similar reproductive physiology to humans, when comparing for example with rabbits, and parturitions are commonly associated with a large foetus, pelvic organ prolapse and dystocia. Moreover, the connective tissue anatomy of the sheep’s pelvic floor is similar to that of humans, thus allowing for more translatable results to human medicine [159].

Studies using goat are at fewer extent to the authors’ knowledge, but recently a group has successfully applied a goat animal model of mastitis with the application of MSCs [160].

### 2.9. Mesenchymal Stem Cells and Small Ruminants

Despite the indisputable importance and relevance of small ruminants as models of animal experimentation, the available data on MSCs from these species is very little when compared to the data accessible for humans or rodent cells. The main limitations are related to the well-known low availability of species-specific or cross-linking antibodies for veterinary species and to discussions related to the MSC definition and nomenclature [161,162]. In addition, there seem to be clear differences in terms of culture conditions and expression of stemness markers that do not allow for a direct translation from the protocols pre-established for humans or rodents to ovine or goat-derived cells [163].

Sheep MSCs (oMSCs) are, like human ones, characterized according to the International Society for Cellular Therapy that include the ability of plastic-adherence, expression of well-defined markers and capacity for tridifferentiation [164]. In general they express surface markers that adapt to the characterization of the ISCT, but some variations in specific markers have already been identified [163]. It is believed that these variations may be related to different sheep breeds, tissues, collection methods, detachment agents used, immunophenotypic variations throughout cell culture and to types of antibodies used to identify the markers [77]. These variations still restrict the broader use of oMSCs therapeutically, and further studies are needed to develop and establish definitive culture and characterization procedures. The proliferation capacity tends to decrease for passages greater than 6 [165], but it can also be affected by variations in culture techniques such as the concentration of fetal bovine serum [166] and tissues where the isolation took place [77]. In vitro culture can alter some of the cell characteristics and plasticity [167]. Karyotypes remain stable up to 20 passages [168] and cells can have steady heredity and viability up to 48 passages, with less stability recorded in cells harvested from fetuses [169]. Genetic instability begins to be seen after extensive culturing, probably related to a shortening of telomeres and aging [73]. The tridifferentiation achieved from different types of chemical compounds allows osteodifferentiation, chondrodifferentiation and adipodifferentiation [170]. Variations in culture conditions and in the microenvironment affect cell differentiation, for example with hypoxia promoting MSCs chondrogenesis [171]. In addition to classic tridifferentiation, other differentiations have been achieved, such as primordial germ cells, cardiomyocytes, endothelial cells or hepatocytes [77]. Other factors such as feeding, age and diseases status of donor animals can influence the characteristics of oMSCs [172] and their differentiation capacity, but cryopreservation and thawing do not appear to influence cell proliferation rates [173].

The exploitation of goat mesenchymal stem cells (gMSCs) in regenerative medicine is still very limited compared with other species, and the understanding of its characteristics and ideal culture conditions are still far from perfect [174]. Studies carried out to date have shown that gMSCs can proliferate up to 20 passages without evident changes, although for higher passages there is an increase in the population doubling time [175]. In addition, MSCs originating from reproductive organs have different population doubling time depending on the stage of the reproductive cycle at collection [176]. Cell proliferation is thought to be dependent on the tissue of origin, with some niches associated with more deleterious effects [177]. Epigenetic changes can also affect their ability for self-renewal [178]. Cellular proliferation of gMSCs is improved through culture with higher concentrations of FBS [179]. As in the case of oMSCs, gMSCs are also characterized according to ISCT criteria. Most cells have the expected spindle shaped fibroblast morphology, although a morphological variety can be observed. Typical markers of MSCs are expressed by gMSCs in general, although with varying intensity. These variations may originate from the collected tissue, collection method, culture conditions, detaching agents and antibodies used for identification, creating the usual problems described in other veterinary species [174,179]. Regarding the differentiation capacity, gMSCs are able to follow the classic tridifferentiation, but under specific conditions they also differentiate in other lines such as neurogenic [180], myogenic [181], epithelial [182] and germ cell-like [183]. After cryopreservation, gMSCs seem to maintain their viability and general characteristics, and even continuing to express surface and pluripotency markers [184].

## 3. Conclusions

In conclusion, the ovine and goat animal models are reliable and appropriate candidates for various disease models, relevant for a translation to human medicine. Both are inexpensive, available in large numbers and require only with simple housing conditions. They are easily handled and docile animals, with low maintenance and feeding costs. Moreover, compared with other domestic animals, like dogs and non-human primates, sheep and goats are ethically and socially more accepted in research. Various groups have been successfully applying these animal models in a wide variety of systems and specific studies, although in most cases the knowledge acquired is still not always translatable and may require more extensive and targeted research. A wide range of future work with these species can be foreseen, which will allow small ruminants to be used as large and translational animal models in regenerative medicine, thus advancing research from animals to human medicine.

## Figures and Tables

**Figure 1 biology-10-00249-f001:**
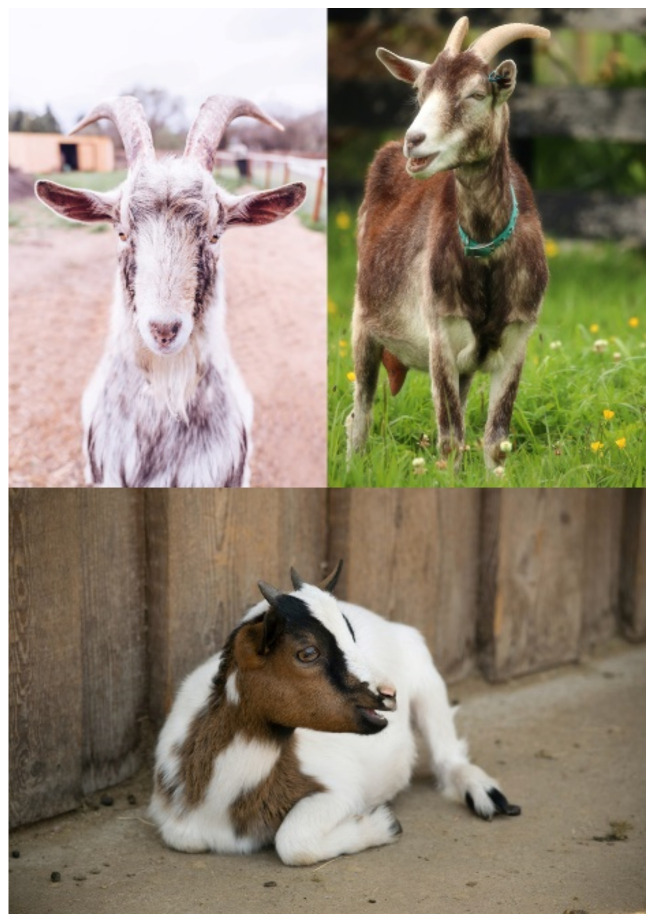
Adult animals (upper panels) and goat (lower panel) of the species *Capra aegagrus hircus.*

**Figure 2 biology-10-00249-f002:**
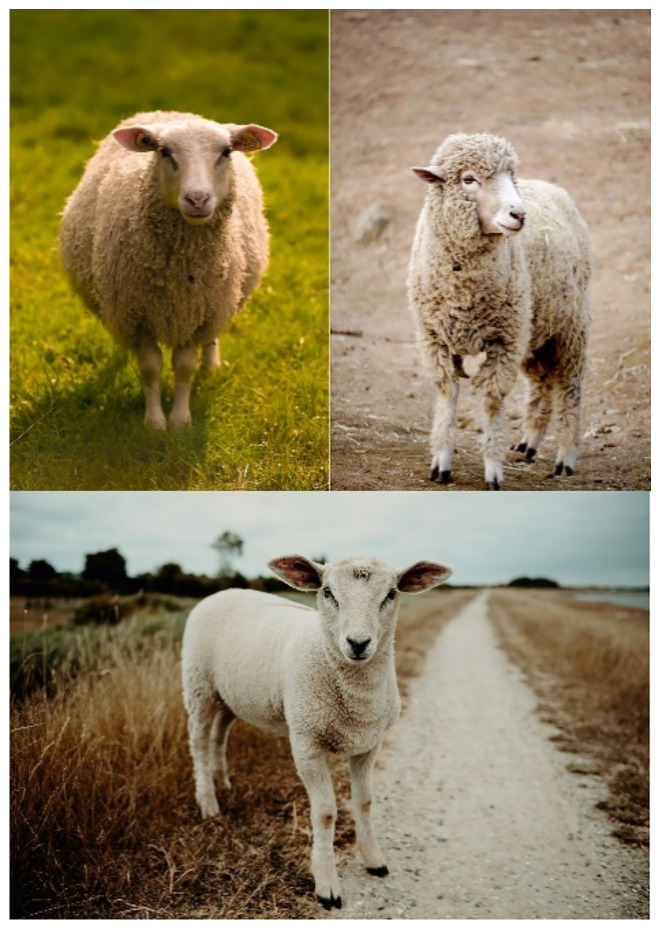
Adult animals (upper panels) and lamb (lower panel) of the species *Ovis aries*.

**Table 1 biology-10-00249-t001:** Normal values of physiological parameters in adult individuals of the species *Ovis aries* (Sheep) and *Capra aegagrus hircus* (Goat) (adapted from [1]).

Parameter/Species	Sheep	Goat
Chromosome number	54	60
Body temperature (°C)	39–40	38.5–39.5
Heart rate (beats/min)	75 (60–120)	85 (70–110)
Respiration rate adult (breaths/min)	36 (12–72)	28 (15–40)
Life span (years)	10–15	8–12 years
Body weights (lbs)	20 to 100 kg	45 to 70 kg
Permanent dental formula	2 (I 0/3 C 0/1 P 3/3 M 3/3) = 32	

**Table 2 biology-10-00249-t002:** Normal values of physiological parameters in young sheep (lambs) (species *Ovis aries*) and goats (kids) (species *Capra aegagrus hircus*) (adapted from [1]).

Parameter/Species	Lambs	Kids
Body temperature (°C)	39.5–40.5	39–40.5
Heart rate (beats/min)	140 (120–160)	140 (120–160)
Respiration rate adult (breaths/min)	50 (30–70)	50 (40–65)
Body weights at Birth (lbs)	1 to 4 Kg	1 to 4 Kg
Deciduos dental formula	2 (Di 0/3 Dc 0/1 Dp 3/3) = 20	

**Table 3 biology-10-00249-t003:** Normal values of reproductive parameters in adults of *Ovis aries* (Sheep) and *Capra aegagrus hircus* (goat). (adapted from [1]).

Reproductive Parameters/Species	Sheep	Goat
Age at puberty (months)	7–8	4–8
Cycle type	Seasonally polyestrus	
Duration of cycle (days)	14–19	18–24
Length of estrus (hours)	24–30	24–96
Gestation (days)	147–150	144–155

## Data Availability

Data sharing not applicable.

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
