# Peer review of "Small Ruminants and Its Use in Regenerative Medicine: Recent Works and Future Perspectives"

_biology, 2021, doi:10.3390/biology10030249_

Round 1

Reviewer 1 Report

could also reference this one:

J Pain Res. 2018; 11: 1147–1162. Published online 2018 Jun 15. doi: 10.2147/JPR.S139843 PMCID: PMC6007193 PMID: 29942150

Ovine model of neuropathic pain for assessing mechanisms of spinal cord stimulation therapy via dorsal horn recordings, von Frey filaments, and gait analysis

Chandan G Reddy,1 John W Miller,1 Kingsley O Abode-Iyamah,1 Sina Safayi,2 Saul Wilson,1 Brian D Dalm,1 Douglas C Fredericks,3 George T Gillies,4 Matthew A Howard, III,1 and Timothy J Brennan5

Author Response

Answer to reviewer 1

Dear reviewer 1:

Thank you very much for the feedback on this review phase, and also for the suggestions made, which received the best attention from us.

As suggested, the following reference was introduced in the manuscript (Reference 52):

  • Reddy CG, Miller JW, Abode-Iyamah KO, Safayi S, Wilson S, Dalm BD, et al. Ovine model of neuropathic pain for assessing mechanisms of spinal cord stimulation therapy via dorsal horn recordings, von Frey filaments, and gait analysis. J Pain Res. 2018, 11:1147-62. doi:10.2147/JPR.S139843

Reviewer 2 Report

Solving a medical problem may need to be studied from different levels and using different species animal models. Mice, rats, and fishes are the most popular used laboratory animal models in medicine, there is no doubt that goats and sheep are useful human disease models.

However, some issues should be concerned

  1. It is not suggested that just because there are several publications, the importance of small ruminants is overextended.
  2. The main application areas that mice and rats cannot replace should be summarized, and the necessity of small ruminants should be emphasized, rather than covering every research area.
  3. This review should summarize the limitations of the small ruminants model, such as: 1) unsatisfactory genetic and microbial control, no inbred and outbred strains; 2) difficult genetically modification, rare models; 3) larger body than conventional laboratory animals, and high cost of use; 4) herbivorous, different from humans; 5) lack of suitable antibodies, etc.
  4. Please explain why the goat application is narrower than sheep.
  5. The introduction is too long, short it; photos are not necessary.

Author Response

Answer to Reviewer 2

Dear Reviewer 2:

The authors would like to acknowledge the Reviewer’s suggestion, which received the best attention from us. The changes made to the document are described below. All changes introduced and highlighted text segments appear in the final document highlighted in yellow.

  • It is not suggested that just because there are several publications, the importance of small ruminants is overextended.

The following text segment was introduced in the document (Line 38-44):

Although the scientific literature presents a multitude of works and projects involving small ruminants in all fields of research, the most significant advances and recurrent use of these animal models have been hampered by a difficulty in establishing standardized models of injury, disease and therapeutic protocols. There is a great variability in terms of the selected breeds, surgical interventions performed and treatments instituted that make it difficult to compare results between different works and more universal and unequivocal conclusions.”

  • The main application areas that mice and rats cannot replace should be summarized, and the necessity of small ruminants should be emphasized, rather than covering every research area:

The advantages of using small ruminants to replace other models in certain research fields have been explained throughout the text in different passages, such as:

“…the anatomical constitution of the subchondral bones and cartilage has have more similarities with humans than other species such as rodents, dogs…” (Lines 107-108)

“…the sheep brain is identical to the human both in terms of dimensions and in the structural organization and location of the cerebral striatal nuclei and cortex. The differences with classic models such as rodents are much more marked.” (Lines 241-244)

“…the anatomy and physiology of the respiratory system in this species is more similar to that of humans than rodents.” (Lines 314-316)

In addition, the following segment was also introduced in the final document (Lines 46-55):

When compared to rodents, the animal models most commonly used in any field of scientific research, small ruminants cannot compete with the fact that these animals breed rapidly and easily, allowing to obtain entire generations of identical and genetically modified animals adapted to certain phenotypic profiles. The quantity and variety of reagents available to study their biology, genetic and immunogenic characteristics is also much higher. However, rodents become less desirable models in assays where thorough surgical interventions or large sample collections are required, and in these cases the great anatomical, physiological and immunological differences make it preferable to use models that are more complex and phylogenetically closer to humans, such as sheep or goats.

  • This review should summarize the limitations of the small ruminants model, such as:
    • Unsatisfactory genetic and microbial control, no inbred and outbred strains:

The following segments were introduced in the final document:

Traditionally, genetic selection in small ruminants was aimed at obtaining populations of animals with better productive performance, but it was essentially carried out by producers and hampered by small animal populations, planning and organization problems, poor animal identification, poor performance and pedigree registration. More recently, however, genetic selection techniques and new molecular tools have become more accessible and standardized and have also started to be applied to improve functional traits such as reproductive performance and disease resistance, allowing the selection of more targeted strains and for scientific research. This genetic selection is essentially based on cross- or multi-breed populations” (Lines 62-70).

Due to its gastrointestinal characteristics and diet, feeding has an essential role in these species and a direct outcome in their immune response and microbiological control. Diet must always be well planned and adapted to the species, and to stressors acting on animals during studies where goats or sheep are involved” (Lines 207-210).

  • Difficult genetically modification, rare models:

The following segments were introduced in the final document:

In the last decades, sheep and goats have also contributed to the development of genetic modification technologies. Although they have been used mainly in the field of genetic engineering to improve their characteristics and increase the production of milk, meat or wool, recently they have also been used as genetic models in biotechnology. The development of new genetic editing technologies that have made this technique more accessible and cheaper has further increased the interest in using these species in genetic bioengineering studies.” (Lines 70-76)

The first drug originated from genetically modified animals and approved by the Euro-pean and United States drug agencies was precisely an anticoagulant drug produced from the milk of transgenic goats (ATryn®)” (Lines 127-129)

Additionally, the following segment was already in the original document:

“The use of transgenic goats to produce a wide variety of biologically active recombinant proteins and antibodies is also common. The desired proteins are collected and isolated from milk, and as such the ideal transgenic animal is one that produces large quantities of milk and has relatively short generation times, and the met both these criteria.” (Lines 122-127).

  • Larger body than conventional laboratory animals, and high cost of use:

The following passages can be found in the original document concerning the dimensions of small ruminants, demonstrating their advantages as ruminants smaller than cattle, but larger than laboratory animals:

The main advantages associated to the use of these animals in research works are related to their anatomy and dimensions, larger than conventional laboratory animals, but very similar to those of humans in most physiological systems…” (Line 16-19)

Their benefits in research, for example in comparison with cattle, are instantly related to its smaller dimensions…” (Lines 44-46)

In addition, since their body dimensions are relatively large, they have also been used in studies with implantation of grafts…” (Lines 111-112)

Their size is similar to those of the humans, allowing good reproducibility in both surgical interventions, sample collection and obtainment of imaging information.” (Lines 163-164)

Regarding the cost of obtaining and maintaining these models, the authors consider it to be low, which is precisely one of the great advantages that justifies their choice. Several passages have been included to reinforce this information:

“…in addition to their low maintenance and feeding costs…” (Line 19)

“… ease of handling and low maintenance costs in reduced areas.” (Line 46)

“… They are easily available animals, with low feeding and maintenance costs and…” (Line 160)

  • Herbivorous, different from humans:

The following segments in the original document include the desired information:

From a dietary standpoint, they are herbivorous animals, and metabolically their only source of glucose is gluconeogenesis” (Lines 82-83)

“Despite their advantages as animal models for most systems, the use of small ruminants as models of gastrointestinal diseases is limited due to their four-stomach system, which fundamentally alters the bioavailability and effectiveness of drugs administered orally.” (Lines 198-201)

  • Lack of suitable antibodies, etc.

The following segments can be found in the original document:

“The main limitations are related to the well-known low availability of species-specific or cross-linking antibodies for veterinary species” (Lines 496-498)

“…immunophenotypic variations throughout cell culture and to types of antibodies used to identify the markers” (Lines 509-510)

“…and antibodies used for its identification, creating the usual problems described in other veterinary species” (Lines 543-544)

  • Please explain why the goat application is narrower than sheep:

The following segments was introduced in the final document (Lines 134-142):

Despite its increasingly widespread use, proportionally the goat continues to be used less frequently than the sheep. This fact can be related to their behavioral characteristics. Because it is a much more interactive, dynamic and social species, goats establish effective connections with humans much more easily than sheep, also raising more ethical questions when selecting these animals for the interventions included in the research work. In addition, their inquisitive, exploratory and less gregarious behavior makes them much more likely to escape and less suitable for assays that require long-term restrictive confinements. Finally, the fact that the genome of goats, unlike sheep, has not yet been fully and unambiguously sequenced, limits the use of this species for genetic studies.”

  • The introduction is too long, short it; photos are not necessary:

The introduction was modified, and some new information was added according to the reviewers' proposals, making it more balanced in terms of content and extension.

Regarding the figures, the authors would like to keep them. Although they are not key images for understanding the text, they can make the document less monotonous and easier to read, and a text without images can make it less appealing.

Reviewer 3 Report

This paper is an interesting review of the literature on the use of sheep and goats in research in regenerative medicine and its potential and actual application to humans.  It is a comprehensive review quoting over 170 references and by and large is well written and clearly laid out .  However, there are several parts where the English can be improved, and the text can be made easier to follow.  I started to indicate this (see the accompanying comments) but then converted the pdf file sent to me to review to a word.doc and the continued to edit and tracked the changes.  I hope that you will be able to use this document but it has not retained the spacing and line numbers accurately.

Current title: Small ruminants and its use in Regenerative Medicine: recent works and future perspectives.

Suggest change to: The use sheep and goats in Regenerative Medicine: recent work and future perspectives.

Line 36: clinics

Line 37: Their not its

Line 31: give some examples of small ruminants early on (e.g. mainly sheep and goats)

Line 46: omit ‘and’

Line 48: rechew not rechewing

Line 50: digits not fingers

Line 69: for not of

Line 75: have not has

Line 83: fainting not faiting

Line 93: and the goat satisfies both these criteria

Line 109: “… is more traditionally used as an animal model

Line 114: more easily accepted

Line 116: Their size rather than its dimensions

Line 121: age not ages

Line 124: such as with other large animal models e.g. cattle (horses are rarely used in medical research)

Line 126: air flow not flows

Tables 1: Capitalisation of Sheep and Goat

Tables 2: Table 2. – Normal values of physiological parameters in young sheep (lambs) (species Ovis aries) and goats (kids) (species Capra aegagrus hircus).

Change titles to Lambs and Kids

Tables 3: Table 3. – Normal reproductive parameters in adults of Ovis aries (Sheep) 150 and Capra aegagrus hircus(Goat).

Capitalisation of Sheep and Goat

All tables omit species in title line just Parameter or Criteria

Line 152: their not its

Line 155: it is also difficult to fast ruminants as their rumen can provide nutrients and ‘water/food’ for several days

Line 191: An ovine transgenic Huntington’s transgenic disease model was already established (52). This sentence does not seem to be related to the sentences either side of it??

Line 199: What are hypoxic-schematic lesions?

Line 203: what is tip?  Tissue transplant/implant

Line 232: intraperitoneal MSCs – are you sure?  Could it be intrapericardial?

Line 356-358 et seq: The goat was used as a model for studying the reconstruction of the corneal epithelium through the use of corneal limbal stem cells, and reconstruction of the damaged corneal surface was observed, probably by inhibition of inflammation related angiogenesis (124)   

This sentence needs to be rephrased

Lines 383-384: Sentences needs clarification as critical and non-critical refer to size, bone and position.

For the whole of my review and other comments please see attached tracked docx file peer-review-10993766.v1.docx

Author Response

Answer to Reviewer 3

Dear Reviewer 3:

The authors would like to acknowledge the Reviewer’s suggestion, which received the best attention from us.

All proposed changes were applied along the manuscript, in line with the reviewer’s comments. The changes can be identified in the manuscript behind track-changes. The track-changes related to the amendments proposed by the reviewer 3 appear in the text without a highlight in yellow. This highlight is related to the changes made after the reviewer's 2 suggestions. However, the adaptations made in the sequence of the proposals of the two reviewers may, in some situations, overlap.

Further notes:

  • Line 252: Hypoxia-ischaemic-injuries are considered by the authors as those induced by blood supply deprivation and consequent inadequate oxygenation of the impaired tissues, thus leading to local ischaemia.
  • Line 255: Author misspelled “type” as “tip” – corrected manuscript: “cell type”
  • Line 287: The authors doubled checked the referenced article (Gugjoo MB. Mesenchymal stem cell research in sheep: Current status and future prospects. Small Ruminant Research. 2018; 169:46-56) and confirmed the MSCs administration route as: endomyocardial, intracoronary and intraperitoneal.
